# Which Coauthor Should I Nominate in My 99 ICLR Submissions? A Mathematical Analysis of the ICLR 2026 Reciprocal Reviewer Nomination Policy

## Abstract

The rapid growth of AI conference submissions has created an overwhelming reviewing burden. To alleviate this, recent venues such as ICLR 2026 introduced a reviewer nomination policy: each submission must nominate one of its authors as a reviewer, and any paper nominating an irresponsible reviewer is desk-rejected. We study this new policy from the perspective of author welfare. Assuming each author carries a probability of being irresponsible, we ask: how can authors (or automated systems) nominate reviewers to minimize the risk of desk rejections? We formalize and analyze three variants of the desk-rejection risk minimization problem. The basic problem, which minimizes expected desk rejections, is solved optimally by a simple greedy algorithm. We then introduce hard and soft nomination limit variants that constrain how many papers may nominate the same author, preventing widespread failures if one author is irresponsible. These formulations connect to classical optimization frameworks, including minimum-cost flow and linear programming, allowing us to design efficient, principled nomination strategies. Our results provide the first theoretical study for reviewer nomination policies, offering both conceptual insights and practical directions for authors to wisely choose which co-author should serve as the nominated reciprocal reviewer.

## 1 Introduction

Artificial Intelligence (AI) has developed at an unprecedented speed and has been applied on an unprecedented scale. A key driving force behind this rapid progress is the role of top AI conferences, which are held annually and have presented countless groundbreaking works, many of the most influential papers of the 21st century. For example, AlexNet was presented at NeurIPS 2012 (Krizhevsky et al., 2012), the Adam optimizer at ICLR 2015 (Kingma & Ba, 2015), ResNet at CVPR 2016 (He et al., 2016), and Transformers at NeurIPS 2017 (Vaswani et al., 2017). These breakthroughs have made AI conferences essential engines of scientific discovery in AI, making them highly impactful and globally important.

However, despite their impact, the dramatic growth in submissions to these conferences has raised serious concerns about the overwhelming reviewing workload (Cao et al., 2025; Li et al., 2025; Allen-Zhu & Xu, 2025). To cope with this challenge, some conferences have begun exploring new policies that distribute reviewing responsibilities by making authors serve as mandatory reviewers (Committees, 2025a;b; 2026b). A notable recent example is the reviewer nomination policy introduced at ICLR 2026 (Figure 1), which requires each submitted paper to nominate one of its authors as a reviewer (Committees, 2026a). If a nominated reviewer is later judged as an irresponsible reviewer, the paper that nominated them is desk-rejected. While this mechanism aims to encourage responsible reviewing, it also raises new concerns about how the nomination choices of authors can directly affect the fate of their submitted papers.

In this work, we study this timely and important problem from the perspective of author welfare. We aim to answer the following research question:

**Question 1.** *Under reviewer nomination policies (like ICLR 2026), how can an author nominate reciprocal reviewers in their submissions wisely to minimize the risk of desk rejections caused by coauthors' irresponsible reviews?*

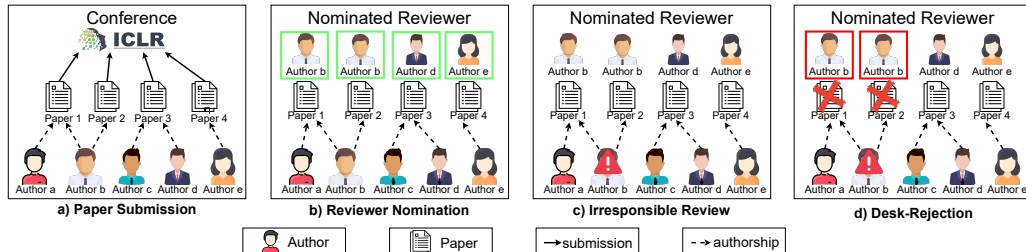

Figure 1: **Irresponsible-revew-related desk-rejection.**

Specifically, we assume that each author has a certain probability of being irresponsible, and thus potentially causing desk rejections for any papers nominating them. To investigate this, we formalize three core problems. The first (Definition 3.1) focuses purely on minimizing expected desk rejections and can be solved optimally by a simple greedy algorithm. The second and the third (Definition 3.4 and Definition 3.9) introduce a constraint on how many papers may nominate the same author, ensuring robust worst-case performance by preventing any single author's irresponsibility from affecting too many papers. Our formulations provide principled algorithmic foundations for reviewer nomination, enabling strategies that significantly improve author welfare in this new policy regime.

Our contributions are summarized as follows:

- **Novel Problem Formulations.** We introduce and formalize three variants of the desk-rejection risk minimization problem, capturing how reviewer nomination strategies interact with desk-rejection risks under new conference policies. Specifically, we study: (i) the basic problem without limits (Definition 3.1), (ii) the problem with hard per-author nomination limits (Definition 3.4), and (iii) the problem with soft per-author nomination limits (Definition 3.9). These are the first formulations to analyze reviewer nomination policies systematically from the authors' perspective.

- **Solution to the Original Problem (Definition 3.1).** In Proposition 3.3, we show that the basic problem is separable across papers and can be solved optimally by a simple greedy algorithm in $O(\mathrm{nnz}(a))$ time, where $\mathrm{nnz}(a)$ is the number of author-paper incidences. This provides a clean baseline and highlights the limitations of only minimizing expected risks.

- **Solution to the Hard Nomination Limit Problem (Definition 3.4).** We prove that the hard limit formulation cannot be solved by naive greedy or random strategies, and that its LP relaxation may yield fractional solutions. Importantly, in Theorem 4.3, we then establish an exact equivalence between this problem and the classical minimum-cost flow problem, ensuring access to modern minimum-cost flow algorithms that has optimality guarantee and high efficiency.

- **Solution to the Soft Nomination Limit Problem (Definition 3.9).** We show that the soft limit formulation is convex but non-smooth, making direct optimization difficult. To address this, in Theorem 4.8, we design an equivalent linear program via auxiliary variables, enabling the use of efficient LP solvers. We further provide a rounding scheme that converts fractional LP solutions into integral assignments while preserving feasibility.

**Roadmap.** In Section 2, we review the relevant works of this paper. In Section 3, we formulate three varuiants of the desk-rejection risk minimization problem. In Section 4, we present our main results. In Section 5, we conclude our paper.

## 2 RELATED WORKS

**Optimization-based reviewer assignment.** The rapid growth in submissions to major conferences has brought increasing attention from the computer science community to improving reviewer assignment, addressing challenges such as bias (Tomkins et al., 2017; Stelmakh et al., 2019a;b), mis-

calibration (Flach et al., 2010; Roos et al., 2011; Ge et al., 2013; Wang & Shah, 2019), personalism (Noothigattu et al., 2021), conflicts of interest (Balietti et al., 2016; Xu et al., 2019), and reviewer-author interactions (Miyao, 2019). To address these challenges, one line of work focuses on bidding mechanisms (Shah et al., 2018; Fiez et al., 2020), which allow reviewers to express preferences over papers.

Another central approach is optimization, which formalizes reviewer assignment as an optimization problem with objectives such as maximizing similarity scores (Cook et al., 2005; Long et al., 2013; Li & Hou, 2016; Aksoy et al., 2023), improving topic coverage (Karimzadehgan & Zhai, 2009; Tayal et al., 2014; Kale et al., 2015), or enhancing fairness (Garg et al., 2010; Kobren et al., 2019; Payan & Zick, 2022). Beyond these objectives, specialized strategies have been proposed for preventing cycles and loops (Guo et al., 2018; Littman, 2021; Leyton-Brown et al., 2024), avoiding torpedo reviewing (Alon et al., 2011; Jecmen et al., 2020; Dhull et al., 2022), and handling conflicts of interest (Merelo-Guervós & Castillo-Valdivieso, 2004; Yan et al., 2019; Pradhan et al., 2020). Other work studies problem formulations, such as grouping strategies (Xu et al., 2010; Wang et al., 2013; Daş & Göçken, 2014) and two-stage reviewing (Leyton-Brown et al., 2024; Jecmen et al., 2022). While these approaches improve aspects such as expertise matching and fairness, in this paper we study a different problem for the first time: reducing desk-rejection risk for authors caused by irresponsible reviewing, which addresses a new policy in leading AI conferences (e.g., ICLR 2026).

**Desk-rejection policies.** A number of different desk-rejection policies have been implemented to decrease the reviewer load in the peer-review process (Ansell & Samuels, 2021). Among these rules, the most widely used is rejecting papers that violate anonymity requirements (Jefferson et al., 2002; Tennant, 2018), which is crucial for ensuring that reviewers from different institutions, career stages, and interests can provide unbiased evaluations. Another common policy targets duplicate and dual submissions (Stone, 2003; Leopold, 2013), helping reduce redundant reviewer effort. Plagiarism (King & ChatGPT, 2023; Elali & Rachid, 2023) is also a key reason for desk-rejection, as it violates academic integrity, infringes on intellectual property, and undermines the credibility of scientific contributions.

To address the rapid growth of AI conference submissions, additional desk-rejection policies have been developed (Leyton-Brown et al., 2024). For example, IJCAI 2020 (Committees, 2020a) and NeurIPS 2020 (Committees, 2020b) adopted a fast-rejection method, allowing area chairs to make desk-rejection decisions based on a quick review of the abstract and main content. Although designed to reduce reviewer workload, this approach introduces unreliability and can misjudge promising work, leading to inappropriate desk-rejections, and thus is rarely adopted. A more widely used strategy is author-level submission limits, where papers are desk-rejected if they include authors who exceed a submission cap. Several conferences have implemented this submission-limit policy, and recent studies examine its fairness and methods for minimizing desk-rejection under this rule (Cao et al., 2025; Li et al., 2025). Recently, ICLR (Committees, 2026a) announced a new desk-rejection policy: a paper will be rejected if the nominated reviewer is deemed irresponsible. Similar policies can also be found in KDD (Committees, 2026b) and NeurIPS (Committees, 2025c). Our work provides a thorough analysis of this new type of desk-rejection policy.

## 3 PROBLEM FORMULATION

In Section 3.1, we present the basic desk-rejection risk minimization problem. In Section 3.2, we extend it with a hard author nomination limit. In Section 3.3, we consider a soft author nomination limit extension of the basic problem.

**Notations.** We use $[n] := \{1, 2, \ldots, n\}$ to denote a set of consecutive positive integers. We use $\mathbb{Z}$ to denote the set of all integers. Let $\mathrm{nnz}(A)$ be the number of non-zero entries in matrix $A$. We use $\mathbf{1}_n$ to denote the $n$-dimensional column vector with all entries equal to 1, and $\mathbf{0}_n$ to denote the $n$-dimensional column vector with all entries equal to 0.

### 3.1 DESK-REJECTION RISK MINIMIZATION

As shown in Figure 1, the ICLR 2026 policy requires each paper to nominate at least one of its authors as a reciprocal reviewer. If a nominated reviewer behaves irresponsibly, then every paper that nominated this author is desk-rejected. From an author's perspective, this introduces a strategic

risk: when submitting multiple papers, the choice of which co-authors to nominate directly affects the probability that some papers will be desk-rejected. This risk is further amplified in recent years, as authors tend to submit more papers and many submissions (e.g., large-scale LLM papers) involve long author lists.

This motivates the following desk-rejection risk minimization problem: authors must carefully choose their nominations across all papers to reduce the probability of desk-rejection caused by irresponsible reviewers. We model this as an integer program.

**Definition 3.1** (Desk-rejection risk minimization, integer program). *Let $n$ denote the number of papers and $m$ the number of authors. Let $a \in \{0,1\}^{n \times m}$ be the authorship matrix, where $a_{i,j} = 1$ if paper $i$ includes author $j$, and $a_{i,j} = 0$ otherwise. Each author $j \in [m]$ is associated with a probability $p_j \in [0,1]$ of being an irresponsible reviewer, with $p_j = 1$ indicating the author is always irresponsible and $p_j = 0$ indicating the author is always responsible.*

*The objective is to minimize the expected number of desk-rejected papers by solving:*

$$\min_{x \in \{0,1\}^{n \times m}} \sum_{i=1}^{n} \sum_{j=1}^{m} x_{i,j} p_j$$

$$\text{s.t. } \sum_{j=1}^{m} a_{i,j} x_{i,j} = 1, \forall i \in [n]$$

*where $x_{i,j} = 1$ indicates that paper $i$ nominates author $j$ as its reciprocal reviewer.*

In the definition above, the constraint $\sum_{j=1}^{m} a_{i,j} x_{i,j} = 1$ ensures that each paper nominates exactly one reciprocal reviewer. Since nominating more authors only increases the risk of desk-rejection without benefit, this formulation naturally restricts each paper to a single nomination.

**Remark 3.2** (Difference from reviewer assignment problems). *The desk-rejection risk minimization problem in Definition 3.1 fundamentally differs from prior work on reviewer-paper matching systems (Yan et al., 2019; Pradhan et al., 2020; Fiez et al., 2020), which focus on expertise matching or conflict of interest avoidance. Our formulation instead takes the author's perspective, aiming to minimize the risk that a nomination leads to desk-rejection due to irresponsible review behaviors.*

Despite its novelty and importance, this formulation only considers the expected number of desk-rejections, while ignoring worst-case risks (e.g., if the least-risk author still turns out irresponsible). Moreover, the problem is computationally trivial, as shown below.

**Proposition 3.3** (Optimal greedy solution for Definition 3.1, informal version of Proposition B.1). *There exists a greedy algorithm that, for each paper, selects the co-author with the smallest irresponsibility probability, and this algorithm solves the desk-rejection risk minimization problem in Definition 3.1 in $O(\text{nnz}(a))$ time.*

This problem is not only technically less challenging but leads to an undesirable worst-case scenario: if the author with the smallest $p_j$ turns out to be irresponsible, then a large number of papers may be rejected. This motivates the need for more robust problem formulations.

## 3.2 DESK-REJECTION RISK MINIMIZATION WITH HARD AUTHOR NOMINATION LIMITS

While the basic problem in Section 3.1 captures the expected risk of desk-rejection, it does not prevent cases where the same "most reliable" author is repeatedly nominated across too many papers. In practice, such a strategy is risky: if that author turns out to be irresponsible, then a large fraction of the submissions may be desk-rejected simultaneously. Thus, we introduce a stricter formulation that enforces a hard limit on the number of papers any single author can be nominated for.

**Definition 3.4** (Desk-rejection risk minimization with hard author nomination limits, integer program). *Let $n$ denote the number of papers and $m$ the number of authors. Let $a \in \{0,1\}^{n \times m}$ be the authorship matrix, where $a_{i,j} = 1$ if paper $i$ includes author $j$, and $a_{i,j} = 0$ otherwise. Each author $j \in [m]$ is associated with an irresponsibility probability $p_j \in [0,1]$. Let $b \in \mathbb{N}_+$ denote the maximum number of papers for which an author can be nominated as reviewer.*

*The objective is to minimize the expected number of desk-rejected papers by solving:*

$$\min_{x \in \{0,1\}^{n \times m}} \sum_{i=1}^{n} \sum_{j=1}^{m} x_{i,j} p_j$$

$$\text{s.t.} \sum_{j=1}^{m} a_{i,j} x_{i,j} = 1, \forall i \in [n]$$

$$\sum_{i=1}^{n} a_{i,j} x_{i,j} \leq b, \forall j \in [m]$$

*where $x_{i,j} = 1$ indicates that paper $i$ nominates author $j$ as its reciprocal reviewer.*

This problem strictly generalizes the formulation in Section 3.1 and is technically more challenging: simple random or greedy algorithms no longer guarantee feasibility.

**Remark 3.5** (Degeneration case of the hard nomination limit problem). *When the nomination limit $b$ is sufficiently large (e.g., $b \geq n$), Definition 3.4 reduces to the original problem in Definition 3.1. In this case, the hard limit never works and every feasible solution of the original problem remains feasible under the limit formulation.*

**Proposition 3.6** (Failure of random and greedy algorithms under hard nomination limits, informal version of Proposition B.2). *There exist instances of Definition 3.4 that has feasible solutions, while both the simple random algorithm (Algorithm 3) and the greedy algorithm (Algorithm 4) fail to return a feasible solution.*

The hard-limit formulation is desirable because it balances risk minimization against catastrophic worst-case outcomes. In Section 4.1, we present theoretical results and algorithms for solving this hard-limit problem efficiently.

However, it still suffers from a structural limitation: for certain inputs, no feasible solution exists. We illustrate this with the following statement:

**Fact 3.7** (Existence of infeasible instances). *There exist choices of $n, m, a, b, p$ in Definition 3.4 such that no feasible solution exists.*

*Proof.* Consider any $n \geq 2$ with $b < n$, and let $m = 1$ with $a = \mathbf{1}_n$. Constraint 1 requires $x = \mathbf{1}_n$, i.e., every paper nominates the only author. But then

$$\sum_{i=1}^{n} a_{i,1} x_{i,1} = n > b,$$

violating Constraint 2. Thus, no feasible solution exists. $\square$

This fact can be illustrated by the following concrete example:

**Example 3.8.** *Let $n = 5$, $m = 1$, $a = [1, 1, 1, 1, 1]^{\top}$, and $b = 2$. This corresponds to an author with 5 single-authored papers but a nomination limit of 2. If the author is nominated in only 2 papers, Constraint 1 fails. If the author is nominated in all 5 papers, Constraint 2 fails. Hence, the instance is infeasible.*

To avoid such impractical cases, we next introduce the soft author nomination limit problem. This relaxation enforces nomination limits in expectation while always guaranteeing the existence of a feasible solution.

### 3.3 DESK-REJECTION RISK MINIMIZATION WITH SOFT AUTHOR NOMINATION LIMITS

The hard-limit formulation in Section 3.2 prevents over-reliance on a single "reliable" author but may result in cases that have no feasible solutions, as shown earlier. To overcome this issue, we relax the hard constraint into a penalty term, allowing every instance to admit a solution while still discouraging excessive nominations of the same author. This leads to the soft author nomination limit problem.

**Definition 3.9** (Desk-rejection risk minimization with soft author nomination limits, integer program). *Let $n$ denote the number of papers and $m$ the number of authors. Let $a \in \{0,1\}^{n \times m}$ be the authorship matrix, where $a_{i,j} = 1$ if paper $i$ includes author $j$, and $a_{i,j} = 0$ otherwise. Each author $j \in [m]$ is associated with an irresponsibility probability $p_j \in [0,1]$. Let $b \in \mathbb{N}_+$ be the nomination limit, and $\lambda > 0$ be a regularization parameter controlling the penalty weight.*

*The objective is to minimize the expected number of desk-rejected papers by solving:*

$$\min_{x \in \{0,1\}^{n \times m}} \sum_{i=1}^{n} \sum_{j=1}^{m} x_{i,j} p_j + \lambda \sum_{j=1}^{m} \max\{0, \sum_{i=1}^{n} a_{i,j} x_{i,j} - b\}$$

$$\text{s.t.} \ \sum_{j=1}^{m} a_{i,j} x_{i,j} = 1, \forall i \in [n]$$

In the definition above, the additional term

$$\lambda \sum_{j=1}^{m} \max\{0, \sum_{i=1}^{n} a_{i,j} x_{i,j} - b\}$$

acts as an $\ell_1$ penalty on each author $j \in [m]$, proportional to the number of nominations exceeding the limit $b$. Intuitively, when $\lambda$ is large, this enforces behavior similar to the hard-limit formulation in Definition 3.4, while still guaranteeing feasibility because violations incur a penalty rather than being strictly forbidden.

**Remark 3.10** (Degeneration under large nomination limits). *When the nomination limit $b$ is sufficiently large (e.g., $b \geq n$), Definition 3.9 reduces to the original formulation in Definition 3.1.*

In Section 4.2, we present theoretical results and algorithms for solving this soft-limit problem efficiently.

# 4 MAIN RESULTS

In Section 4.1, we present our theoretical results on solving the hard author nomination limit problem. In Section 4.2, we extend our theoretical results to the soft nomination limits.

## 4.1 SOLVING HARD AUTHOR NOMINATION LIMITS

**Linear Program Relaxation.** To solve the hard author nomination problem in Definition 3.4, a natural first step is to relax the integer program into a linear program:

**Definition 4.1** (Desk-rejection risk minimization with hard author nomination limits, relaxed linear program). *Let $n$ denote the number of papers and $m$ the number of authors. Let $a \in \{0,1\}^{n \times m}$ be the authorship matrix, where $a_{i,j} = 1$ if paper $i$ includes author $j$, and $a_{i,j} = 0$ otherwise. Each author $j \in [m]$ is associated with an irresponsibility probability $p_j \in [0,1]$. Let $b \in \mathbb{N}_+$ denote the maximum number of papers for which an author can be nominated as reviewer.*

*The objective is to minimize the expected number of desk-rejected papers by solving:*

$$\min_{x \in [0,1]^{n \times m}} \sum_{i=1}^{n} \sum_{j=1}^{m} x_{i,j} p_j$$

$$\text{s.t.} \ \sum_{j=1}^{m} a_{i,j} x_{i,j} = 1, \forall i \in [n]$$

$$\sum_{i=1}^{n} a_{i,j} x_{i,j} \leq b, \forall j \in [m]$$

Although linear programs can be solved efficiently, the relaxation may yield fractional solutions that are not valid assignments in the original integer problem. In Proposition 4.2 we showed that such

fractional optima indeed exist. This motivates us to search for an exact combinatorial algorithm with guaranteed integrality.

**Proposition 4.2** (Existence of fractional optima for the relaxed hard nomination problem, informal version of Proposition C.1). *There exists an instance of Definition 4.1 whose optimal solution $x$ is fractional (i.e., there exist indices $i \in [n]$ and $j \in [m]$ such that $x_{i,j} \notin \{0,1\}$).*

**Minimum-Cost Flow Equivalence.** The hard author nomination limit problem can be reformulated as a minimum-cost circulation problem, and hence as a special case of the minimum-cost flow problem. The intuition is straightforward:

- Each paper $i \in [n]$ corresponds to a demand of exactly one unit of flow (since it must nominate exactly one reviewer).
- Each author $j \in [m]$ corresponds to a capacity-limited supply node, where the outgoing flow cannot exceed the nomination limit $b$.
- Edges between authors and papers encode the author-paper matrix (i.e., $a_{i,j} = 1$), with cost equal to the irresponsibility probability $p_j$.
- By routing $n$ units of flow from a source vertex through authors into papers and then to a sink vertex, we enforce both feasibility and the per-paper nomination requirement.

This construction transforms Definition 3.4 into a network flow instance with integral capacities and costs. A key property of minimum-cost flow is that whenever capacities and demands are integral, the problem always admits an optimal integral solution. Thus, unlike the LP relaxation, the flow-based formulation guarantees feasibility and integrality without the need for rounding.

Therefore, we obtain the following result:

**Theorem 4.3** (Equivalence to minimum-cost flow, informal version of Theorem C.2). *The hard author nomination problem in Definition 3.4 is equivalent to a minimum-cost flow problem, and therefore always admits an optimal integral solution whenever a feasible assignment exists.*

The minimum-cost flow problem has been studied extensively, with polynomial-time algorithms (Edmonds & Karp, 1972; Goldberg & Tarjan, 1990; Daitch & Spielman, 2008) available since 1972. Our result in Theorem 4.3 enables the hard-limit problem to be solved not only by these classical algorithms, many of which already have mature off-the-shelf implementations, but also by the most recent state-of-the-art methods.

**Remark 4.4** (Efficiency and practical implications). *Recent state-of-the-art results achieve $\widetilde{O}(N + M^{1.5})$[1] time for minimum-cost flow problems with $N$ vertices and $M$ edges (Brand et al., 2021). For large AI conferences such as ICLR, where both $N$ and $M$ scale on the order of $10^4$ (Li et al., 2025), the hard nomination limit problem in Definition 3.4 can be solved efficiently in practice while remaining exact and optimal integer solutions.*

### 4.2 Solving Soft Author Nomination Limits

**Relaxing the Integer Program.** Hard nomination limits can make some instances infeasible. A natural remedy is to "soften" the limit by penalizing nomination overloads rather than totally forbidding them in the constraints. This preserves feasibility while discouraging solutions that concentrate nominations on a few authors. To solve the soft author nomination limit problem in Definition 3.9, we first relax the original integer program into a continuous form.

**Definition 4.5** (Desk-rejection risk minimization with soft author nomination limits, relaxed problem). *Let $n$ denote the number of papers and $m$ the number of authors. Let $a \in \{0,1\}^{n \times m}$ be the authorship matrix, where $a_{i,j} = 1$ if paper $i$ includes author $j$, and $a_{i,j} = 0$ otherwise. Each author $j \in [m]$ is associated with an irresponsibility probability $p_j \in [0,1]$. Let $b \in \mathbb{N}_+$ be the nomination limit, and $\lambda > 0$ be a regularization parameter controlling the penalty weight.*

*The objective is to minimize the expected number of desk-rejected papers by solving:*

$$\min_{x \in [0,1]^{n \times m}} \sum_{i=1}^{n} \sum_{j=1}^{m} x_{i,j} p_j + \lambda \sum_{j=1}^{m} \max\{0, \sum_{i=1}^{n} a_{i,j} x_{i,j} - b\}$$

---

[1] The $\widetilde{O}$-notation omits the $N^{o(1)}$ and $M^{o(1)}$ factors.

$$\text{s.t. } \sum_{j=1}^{m} a_{i,j} x_{i,j} = 1, \forall i \in [n]$$

**Fact 4.6** (Convexity of Definition 4.5). *Problem 4.5 is a convex optimization problem.*

*Proof.* The objective is a sum of a linear term and a sum of convex functions composed with affine maps. The constraints are linear equalities together with box constraints $0 \le x_{i,j} \le 1, \forall i \in [n], j \in [m]$, so the feasible set is convex. Therefore, the problem is convex, which finishes the proof. $\square$

Although the relaxed soft author nomination limit problem is convex, it is still algorithmically non-straightforward. The second term introduces non-smoothness in the objective, and unlike standard smooth–non-smooth decompositions, it is not separable from the linear part. Combined with the equality constraints, this coupling makes the problem harder to solve directly.

**Linear Program Re-formulation.** To address these challenges, we propose a linear programming reformulation inspired by the epigraph trick (Boyd & Vandenberghe, 2004), which removes non-smoothness through auxiliary variables and facilitates the use of efficient LP solvers with robust theoretical guarantees and practical implementations.

**Definition 4.7** (Desk-rejection risk minimization with soft author nomination limits, linear program). *Let $n$ denote the number of papers and $m$ the number of authors. Let $a \in \{0,1\}^{n \times m}$ be the authorship matrix, where $a_{i,j} = 1$ if paper $i$ includes author $j$, and $a_{i,j} = 0$ otherwise. Each author $j \in [m]$ is associated with an irresponsibility probability $p_j \in [0,1]$. Let $b \in \mathbb{N}_+$ be the nomination limit, and $\lambda > 0$ be a regularization parameter controlling the penalty weight.*

*Let $y \in \mathbb{R}_+^m$ denote the penalty for nominating an author in too many papers. The objective is to minimize the expected number of desk-rejected papers by solving:*

$$\min_{x \in [0,1]^{n \times m}, y \in \mathbb{R}_+^m} \sum_{i=1}^{n} \sum_{j=1}^{m} x_{i,j} p_j + \lambda \sum_{j=1}^{m} y_j$$

$$\text{s.t. } \sum_{j=1}^{m} a_{i,j} x_{i,j} = 1, \forall i \in [n]$$

$$y_j \ge \sum_{i=1}^{n} a_{i,j} x_{i,j} - b, j \in [m]$$

**Theorem 4.8** (Equivalence of the convex and linear program formulations for the soft nomination limit problem, informal version of Theorem C.3). *The relaxed soft author nomination limit problem in Definition 4.5 and its linear program reformulation in Definition 4.7 are equivalent with respect to the assignment variable $x$.*

This LP reformulation enables us to solve the soft author nomination limit problem in Definition 3.9 using modern linear programming solvers, which provide both strong optimality guarantees and high practical efficiency. To assess the computational efficiency of such solvers, we first discuss sparsification and then present complexity considerations.

**Remark 4.9** (Sparsification of decision variables). *In practice, the optimization variables $x_{i,j}$ only need to be maintained for entries with $a_{i,j} = 1$. For all pairs $(i, j) \in [n] \times [m]$ with $a_{i,j} = 0$, we can fix $x_{i,j} = 0$ without affecting feasibility or optimality. Thus, the total number of decision variables reduces from $O(nm)$ to $O(\text{nnz}(a))$.*

**Remark 4.10** (Complexity of solving the LP reformulation). *The time complexity of solving the LP in Definition 4.7 aligns with the performance of modern linear programming methods. For example, using the stochastic central path method (Cohen et al., 2021; Jiang et al., 2021; Dong et al., 2021), the problem can be solved in $\widetilde{O}(K^{2.37} \log(K/\delta))$ time[2], where $K$ is the number of decision variables and $\delta$ is the relative accuracy parameter of a $(1 + \delta)$-approximation guarantee.*

---

[2]The $\widetilde{O}$-notation omits the $K^{o(1)}$ factors.

**Remark 4.11** (Practical implications). *Empirical statistics from recent AI conferences (e.g., ICLR) show that* $\mathrm{nnz}(a)$ *typically scales to the order of* $10^4$ *(Li et al., 2025). At this scale, the LP reformulation can be solved efficiently within available computational resources. This ensures that our formulation is not only theoretically sound but also practical for real-world conference applications.*

**Rounding.** Solving the LP relaxation (Definition 4.7) of the soft author nomination problem (Definition 3.9) provides a (possibly) fractional solution $x \in [0,1]^{n \times m}$. While this fractional solution is optimal in the relaxed space, it is not directly implementable in practice because reviewer nominations must be integral: each paper must nominate exactly one author, not a weighted combination of several authors. Thus, rounding is required to transform the LP's fractional solution into a valid integer solution that respects the feasibility constraints of the original problem.

The rounding algorithm in Algorithm 1 achieves this by selecting, for each paper, the author with the largest fractional assignment and setting the corresponding decision variable to 1, ensuring that every paper nominates exactly one reviewer.

---

**Algorithm 1** Rounding Algorithm for Desk-Rejection Risk Minimization with Soft Author Nomination Limit (Definition 4.7)

---

1: **procedure** ROUNDINGSOFT($a \in \{0,1\}^{n \times m}, x \in [0,1]^{n \times m}, n, m, b \in \mathbb{N}_+, p \in (0,1)^m$)
2: $\quad \widetilde{x} \leftarrow \mathbf{0}_{n \times m}$
3: $\quad$ **for** $i \in [n]$ **do**
4: $\quad\quad j_\star \leftarrow \arg\max_{j \in [m]} x_{i,j}$ $\qquad\qquad\qquad$ ▷ pick the largest fractional value in row $i$
5: $\quad\quad \widetilde{x}_{i,j_\star} \leftarrow 1$
6: $\quad$ **end for**
7: $\quad$ **return** $\widetilde{x}$
8: **end procedure**

---

**Proposition 4.12** (Correctness and efficiency of rounding algorithm). *Algorithm 1 produces a feasible solution* $\widetilde{x} \in \{0,1\}^{n \times m}$ *to the integer soft nomination limit problem in Definition 3.9 in* $O(\mathrm{nnz}(a))$ *time.*

*Proof.* **Part 1. Correctness.** For each paper $i \in [n]$, let $S_i := \{j \in [m] : a_{i,j} = 1\}$ denote its author set. By the constraints of the linear program, we have $x_{i,j} = 0$ for $j \notin S_i$ and $\sum_{j \in S_i} x_{i,j} = 1$. Algorithm 1 selects $j_\star \in \arg\max_{j \in S_i} x_{i,j}$ and sets $\widetilde{x}_{i,j_\star} = 1$ and $\widetilde{x}_{i,j} = 0$ for $j \neq j_\star$. Hence, exactly one author in $S_i$ is chosen and none outside $S_i$, i.e.,

$$\sum_{j=1}^m a_{i,j}\widetilde{x}_{i,j} = 1, \forall i \in [n].$$

This satisfies the per-paper nomination constraint, which is the only constraint in Definition 3.9. Thus, we can conlude that $\widetilde{x}$ is feasible.

**Part 2. Running Time.** For each paper $i$, the algorithm scans $S_i$ once to find $j_\star$, which takes $O(\mathrm{nnz}(a_{i,*}))$ time. Summing over all papers gives total time

$$O(\sum_{i=1}^n \mathrm{nnz}(a_{i,*})) = O(\mathrm{nnz}(a)).$$

Therefore, Algorithm 1 runs in $O(\mathrm{nnz}(a))$ time. $\qquad\qquad\qquad\qquad\qquad\qquad\qquad$ □

## 5 CONCLUSION

This paper provides the first systematic study of reviewer nomination policies from the perspective of author welfare. By formulating the desk-rejection risk minimization problem and analyzing three variants, we establish clean theoretical foundations and efficient algorithms for mitigating nomination-related desk rejections. Our results show that while the basic problem admits a simple greedy solution, the hard and soft nomination limit problems connect naturally to well-studied optimization paradigms such as minimum-cost flow and linear programming, allowing us to leverage decades of algorithmic progress.

ETHIC STATEMENT

This paper does not involve human subjects, personally identifiable data, or sensitive applications. We do not foresee direct ethical risks. We follow the ICLR Code of Ethics and affirm that all aspects of this research comply with the principles of fairness, transparency, and integrity.

REPRODUCIBILITY STATEMENT

We ensure reproducibility of our theoretical results by including all formal assumptions, definitions, and complete proofs in the appendix. The main text states each theorem clearly and refers to the detailed proofs. No external data or software is required.

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

# Appendix

**Roadmap.** In Section A, we explain some background knowledge on network flow problems. In Section B, we show the missing proofs in Section 3. In Section C, we supplement the missing proofs in Section 4. In Section D, we present several useful baseline algorithms for the soft nomination limit problems.

## A BACKGROUNDS ON NETWORK FLOW

We introduce the minimum-cost flow problem in Section A.1, and then introduce the equivalent minimum-cost circulation problem in Section A.2.

### A.1 MINIMUM-COST FLOW PROBLEM

**Definition A.1** (Feasible flow, implicit in page 296 of (Ahuja et al., 1993)). *Let $G := (V, E)$ be a directed graph, where $V := [N]$ is the vertex set and $E := \{(i_1, j_1), (i_2, j_2), \dots, (i_M, j_M)\}$ is the edge set. For each $(i, j) \in E$, we define a capacity $c(i, j) \in \mathbb{Z}_{\geq 0}$, and a cost $w(i, j) \in \mathbb{R}$. Each vertex $v \in V$ is associated with a supply/demand value $b(v) \in \mathbb{Z}$ such that $\sum_{v \in V} b(v) = 0$.*

*We say that a function $f : E \to \mathbb{R}_{\geq 0}$ is a feasible flow if $f$ satisfies:*

- *$0 \leq f(i, j) \leq c(i, j), \forall (i, j) \in E$,*

- *$\sum_{k:(v,k) \in E} f(v, k) - \sum_{k:(k,v) \in E} f(k, v) = b(v), \forall v \in V$.*

**Definition A.2** (Minimum-cost flow problem, implicit in page 296 of (Ahuja et al., 1993)). *Given a feasible flow $f$ as in Definition A.1, the minimum-cost flow problem is to find a feasible flow $f$ that minimizes*

$$\sum_{(i,j) \in E} w(i, j) \cdot f(i, j).$$

### A.2 MINIMUM-COST CIRCULATION PROBLEM

**Definition A.3** (Circulation, implicit in page 194 of (Ahuja et al., 1993)). *Let $G := (V, E)$ be a directed graph, where $V := [N]$ is the vertex set and $E := \{(i_1, j_1), (i_2, j_2), \dots, (i_M, j_M)\}$ is the edge set. For all $(i, j) \in E$, we define a demand $d(i, j) \in \mathbb{Z}_{\geq 0}$ and a capacity $c(i, j) \in \mathbb{Z}_{\geq 0}$ such that $0 \leq d(i, j) \leq c(i, j)$.*

*We say that a function $f : E \to \mathbb{R}_{\geq 0}$ is a circulation if $f$ satisfies:*

- *$d(i, j) \leq f(i, j) \leq c(i, j), \forall (i, j) \in E$,*

- *$\sum_{k:(i,k) \in E} f(i, k) = \sum_{k:(k,i) \in E} f(k, i), \forall i \in [N]$.*

**Definition A.4** (Minimum-cost circulation problem, implicit in page 1 of (Williamson, 2007)). *Given a feasible circulation $f$ as in Definition A.3, the minimum-cost flow problem is to find a circulation $f$ that minimizes*

$$\sum_{(i,j) \in E} w(i, j) \cdot f(i, j).$$

**Lemma A.5** (Equivalence of minimum-cost circulation and minimum-cost flow, Theorem 1 on page 2 of (Williamson, 2007)). *The minimum-cost flow problem in Definition A.2 and the minimum-cost circulation problem in Definition A.4 are equivalent.*

## B MISSING PROOFS IN SECTION 3

We first supplement the proof for Proposition 3.3.

**Proposition B.1** (Optimal greedy solution for Definition Proposition 3.1, formal version of Proposition 3.3)**.** *There exists a greedy algorithm that, for each paper, selects the co-author with the smallest irresponsibility probability, and this algorithm solves the desk-rejection risk minimization problem in Definition 3.1 in $O(\mathrm{nnz}(a))$ time.*

*Proof.* We present the greedy algorithm in Algorithm 2 and finish the proof in two parts.

**Part 1: Optimality.** The objective value for any feasible assignment $x \in \{0,1\}^{n \times m}$ is $\sum_{i=1}^{n} \sum_{j=1}^{m} x_{i,j} p_j$, which is separable across papers. Thus, the choice of a reviewer for paper $i$ does not affect the choices for papers $i+1, i+2, \ldots, n$. Minimizing the overall objective therefore reduces to independently minimizing each paper's contribution.

For a fixed paper $i \in [n]$, the objective is minimized by selecting any author $j \in [m]$ such that $a_{i,j} = 1$ and $p_j$ is minimized. Algorithm 2 implements exactly this strategy: it identifies the set of minimizers

$$S_{\min} = \{j \in S_{\mathrm{all}} : p_j = \min_{k \in S_{\mathrm{all}}} p_k\},$$

and then chooses one $k \in S_{\min}$, setting $x_{i,k} = 1$. Since the problem is separable, independently minimizing each term yields a globally optimal solution.

**Part 2: Time Complexity.** For each paper $i$, let $S_{\mathrm{all}} = \{j \in \mathbb{Z}_+ : a_{i,j} = 1\}$ be the set of its authors. Constructing this set requires $O(\mathrm{nnz}(a_{i,*}))$ time, and computing $\min_{k \in S_{\mathrm{all}}} p_k$ and $S_{\min}$ also takes $O(\mathrm{nnz}(a_{i,*}))$. Therefore, the total time across all $n$ papers is

$$\sum_{i=1}^{n} \mathrm{nnz}(a_{i,*}) = \mathrm{nnz}(a).$$

Thus, the overall running time is $O(\mathrm{nnz}(a))$.

Combining Part 1 and Part 2, we finish the proof. □

---

**Algorithm 2** Simple Greedy Algorithm for the Problem in Definition 3.1

---

1: **procedure** GREEDYASSIGN1($a \in \{0,1\}^{n \times m}$, $p \in (0,1)^m$, $n, m \in \mathbb{Z}_+$)
2:     $x \leftarrow \mathbf{0}_{n \times m}$
3:     **for** $i \in [n]$ **do**                                              ▷ Iterate all the papers
4:         $S_{\mathrm{all}} \leftarrow \{j \in \mathbb{Z}_+ : a_{i,j} = 1\}$        ▷ A set including all the authors for this paper
5:         $S_{\min} \leftarrow \{j \in S_{\mathrm{all}} : p_j = \min_{k \in S_{\mathrm{all}}} p_k\}$        ▷ Keep the most responsible authors
6:         Randomly choose an element $k$ from $S_{\min}$
7:         $x_{i,k} \leftarrow 1$
8:     **end for**
9:     **return** $x$
10: **end procedure**

---

Next, we show the proof for Proposition 3.6.

**Proposition B.2** (Failure of random and greedy algorithms under hard nomination limits, formal version of Proposition 3.6)**.** *There exist instances of Definition 3.4 that has feasible solutions, while both the simple random algorithm (Algorithm 3) and the greedy algorithm (Algorithm 4) fail to return a feasible solution.*

*Proof.* Let's consider the following example: $n = 2, m = 2, a = \begin{bmatrix} 1 & 1 \\ 1 & 0 \end{bmatrix}, b = 1$.

**Part 1. Random Algorithm.** In Algorithm 3, we first enter the for-loop with $i = 1$. At line 5, we get $S_{\mathrm{all}} = \{1, 2\}$, and at line 6 we get $S_{<b} = \{1, 2\}$. Suppose the algorithm randomly selects author 1 at line 11.

Next, we enter the for-loop with $i = 2$. At this point, we have $S_{\text{all}} = \{1\}$ at line 5. However, author 1 already has one nomination, so $\sum_{i=1}^{2} a_{i,1} x_{i,1} + 1 = 2 > b$, which makes $S_{<b} = \emptyset$ at line 6, i.e., $|S_{<b}| = 0$. This triggers the if-branch at line 9, setting the error flag to true.

Thus, the random algorithm cannot produce a valid solution.

**Part 2. Greedy Algorithm.** In Algorithm 4, we first enter the for-loop with $i = 1$. At line 5, we get $S_{\text{all}} = \{1, 2\}$, and at line 6, we get $S_{<b} = \{1, 2\}$. Since both authors are valid and we have $p_1 = p_2$, line 11 returns two equivalently "most responsible" authors, i.e., $S_{\max} = \{1, 2\}$. Suppose line 12 then randomly selects author 1.

Next, we enter the for-loop with $i = 2$. Now $S_{\text{all}} = \{1\}$ at line 5, but author 1 already has one nomination. Hence $\sum_{i=1}^{2} a_{i,1} x_{i,1} + 1 = 2 > b$, which makes $S_{<b} = \emptyset$ at line 6, triggering the if-branch at line 9 and setting the error flag to true.

Thus, the greedy algorithm also fails to produce a valid solution.

Finally, by combining Parts 1-2 of the proof, we can finish the proof. $\qquad \square$

---

**Algorithm 3** Simple Random Algorithm for Desk-Rejection Risk Minimization with Hard Author Nomination Limit Problem in Definition 3.4

---

1: **procedure** RANDASSIGNHARD($a \in \{0,1\}^{n \times m}, n, m, b \in \mathbb{Z}_+$)
2:      $x \leftarrow \mathbf{0}_{n \times m}$
3:      err $\leftarrow$ false             $\triangleright$ Error flag
4:      **for** $i \in [n]$ **do**          $\triangleright$ Iterate all the papers
5:          $S_{\text{all}} \leftarrow \{j \in [m] : a_{i,j} = 1\}$      $\triangleright$ A set including all the authors for this paper
6:          $S_{<b} \leftarrow \{j \in S_{\text{all}} : \sum_{i=1}^{n} a_{i,k} x_{i,k} + 1 \leq b\}$    $\triangleright$ Authors nominated by less than $b$ times
7:          **if** $|S_{<b}| = 0$ **then**         $\triangleright$ Cannot obtain a valid choice of author
8:              Randomly choose an element $k$ from $S_{\text{all}}$
9:              err $\leftarrow$ true
10:         **else**             $\triangleright$ Has a valid choice of author
11:            Randomly choose an element $k$ from $S_{<b}$
12:         **end if**
13:         $x_{i,k} \leftarrow 1$
14:      **end for**
15:      **return** $x$, err
16: **end procedure**

---

**Algorithm 4** Simple Greedy Algorithm for Desk-Rejection Risk Minimization with Hard Author Nomination Limit Problem in Definition 3.4

---

1: **procedure** GREEDYASSIGNHARD($a \in \{0,1\}^{n \times m}, n, m, b \in \mathbb{Z}_+, p \in (0,1)^m$)
2:      $x \leftarrow \mathbf{0}_{n \times m}$
3:      err $\leftarrow$ false             $\triangleright$ Error flag
4:      **for** $i \in [n]$ **do**          $\triangleright$ Iterate all the papers
5:          $S_{\text{all}} \leftarrow \{j \in [m] : a_{i,j} = 1\}$      $\triangleright$ A set including all the authors for this paper
6:          $S_{<b} \leftarrow \{j \in S_{\text{all}} : \sum_{i=1}^{n} a_{i,k} x_{i,k} + 1 \leq b\}$    $\triangleright$ Authors nominated by less than $b$ times
7:          **if** $|S_{<b}| = 0$ **then**         $\triangleright$ Cannot obtain a valid choice of author
8:              Randomly choose an element $k$ from $S_{\text{all}}$
9:              err $\leftarrow$ true
10:         **else**             $\triangleright$ Has a valid choice of author
11:            $S_{\min} \leftarrow \{j \in S_{\text{all}} : p_j = \min_{k \in S_{<b}} p_k\}$    $\triangleright$ Keep the most responsible authors
12:            Randomly choose an element $k$ from $S_{\max}$
13:         **end if**
14:         $x_{i,k} \leftarrow 1$
15:      **end for**
16:      **return** $x$, err
17: **end procedure**

---

## C  MISSING PROOFS IN SECTION 4

We begin by showing the proof for Proposition 4.2.

**Proposition C.1** (Existence of fractional solution in relaxed hard author nomination limit problem, formal version of Proposition 4.2)**.** *There exists a global minimum $x$ of the problem in Definition 4.1 such that $\exists i \in [n], j \in [m], x_{i,j} \notin \{0, 1\}$.*

*Proof.* To establish existence, it suffices to construct an example that yields a fractional optimal solution. Consider the case where $n = m = 2$, $b = 1$, $a = \begin{bmatrix} 1 & 1 \\ 1 & 1 \end{bmatrix}$, and $p = \begin{bmatrix} 1/6 \\ 1/6 \end{bmatrix}$.

The corresponding linear program for Definition 4.1 is:

$$\min_{x} \sum_{i=1}^{2} \sum_{j=1}^{2} \frac{1}{6} x_{i,j}$$

$$\text{s.t.} \sum_{j=1}^{2} x_{i,j} = 1, \quad \forall i \in [2],$$

$$0 \leq x_{i,j} \leq 1, \quad \forall i \in [2], j \in [2],$$

$$\sum_{i=1}^{2} x_{i,j} \leq 1, \quad \forall j \in [2].$$

This program is equivalent to:

$$\min_{x} \frac{1}{6} \cdot (x_{1,1} + x_{1,2} + x_{2,1} + x_{2,2})$$
$$\text{s.t.} \ x_{1,1} + x_{1,2} = 1,$$
$$x_{2,1} + x_{2,2} = 1,$$
$$0 \leq x_{i,j} \leq 1, \quad \forall i \in [2], j \in [2],$$
$$x_{1,1} + x_{2,1} \leq 1,$$
$$x_{1,2} + x_{2,2} \leq 1.$$

Eliminating $x_{1,2}$ and $x_{2,2}$ using the equality constraints, we have:

$$\min_{x_{1,1}, x_{2,1}} \ 0$$
$$\text{s.t.} \ 0 \leq x_{1,1} \leq 1,$$
$$0 \leq x_{2,1} \leq 1,$$
$$x_{1,1} + x_{2,1} \leq 1,$$
$$(1 - x_{1,1}) + (1 - x_{2,1}) \leq 1.$$

Since combining the last two constraints gives $x_{1,1} + x_{2,1} = 1$ and the objective function is a constant, we can conclude that any solution of the form

$$\begin{bmatrix} x_{1,1} & 1 - x_{1,1} \\ 1 - x_{1,1} & x_{1,1} \end{bmatrix}, \forall x_{1,1} \in [0, 1]$$

is a global minimum of the original problem. In particular, any choice with $x_{1,1} \in (0, 1)$ yields a fractional solution such that $\exists i \in [n], j \in [m], x_{i,j} \notin \{0, 1\}$.

This completes the proof. $\square$

Next, we show the proof for Theorem 4.8.

**Theorem C.2** (Equivalence to minimum-cost flow, formal version of Theorem 4.3)**.** *The hard author nomination problem in Definition 3.4 is a minimum-cost flow problem in Definition A.2.*

*Proof.* We construct a network $G = (V, E)$ as follows. The vertex set is $V = [m+n+2]$, consisting of

- Source vertex 1,

- Target vertex 2,

- Author vertices $\{3, 4, \ldots, m + 2\}$,

- Paper vertices $\{m + 3, m + 4, \ldots, m + n + 2\}$.

The edge set $E$ is defined as:

- For each author $j \in [m]$, add an edge $(1, j + 2)$ with $d(1, j + 2) = 0$, $c(1, j + 2) = b$, and $w(1, j + 2) = 0$, encoding that each author can be nominated at most $b$ times.

- For each $(i, j)$ with $i \in [n], j \in [m]$ and $a_{i,j} = 1$, add an edge $(j + 2, i + m + 2)$ with $d(j + 2, i + m + 2) = 0$, $c(j + 2, i + m + 2) = 1$, and $w(j + 2, i + m + 2) = p_j$, encoding the cost of nominating author $j$ for paper $i$.

- For each paper $i \in [n]$, add an edge $(i+m+2, 2)$ with $d(i+m+2, 2) = c(i+m+2, 2) = 1$ and $w(i + m + 2, 2) = 0$, enforcing that each paper must nominate exactly one reviewer.

- Add an edge $(2, 1)$ with $d(2, 1) = 0$, $c(2, 1) = n$, and $w(2, 1) = 0$, ensuring circulation of total flow.

Let $f$ be an optimal circulation of this network. Then the assignment matrix $x \in \{0, 1\}^{n \times m}$ is recovered from the author-paper edges as

$$x_{i,j} = f(j + 2, i + m + 2), \quad \forall i \in [n], j \in [m].$$

This shows that the hard author nomination problem is equivalent to a minimum-cost circulation problem. By Lemma A.5, the minimum-cost circulation and minimum-cost flow problems are equivalent, and thus the hard nomination limit problem is a minimum-cost flow problem.

Thus, we finish the proof. $\qquad\square$

Then, we show the proof for the linear program re-formulation in Theorem 4.8.

**Theorem C.3** (Equivalence of the convex and linear program formulations for the soft nomination limit problem, formal version of Theorem 4.8)**.** *The optimal solution $x_{\mathrm{OPT},1}$ of the problem in Definition 4.5 and the optimal solution $(x_{\mathrm{OPT},2}, y_{\mathrm{OPT}})$ of the problem in Definition 4.7 is the same, i.e., $x_{\mathrm{OPT},1} = x_{\mathrm{OPT},2}$.*

*Proof.* Fix any $j \in [m]$. Consider the global minimum $y_{\mathrm{OPT},j}$ in Definition 4.7:

**Case 1.** If $\sum_{i=1}^{n} a_{i,j} x_{\mathrm{OPT},2,i,j} - b \geq 0$, the constraint $y_j \geq \sum_{i=1}^{n} a_{i,j} x_{\mathrm{OPT},2,i,j} - b$ is active. To minimize the objective, $y_{\mathrm{OPT},j}$ must equal this lower bound. Thus, we have

$$y_{\mathrm{OPT},j} = \sum_{i=1}^{n} a_{i,j} x_{\mathrm{OPT},2,i,j} - b$$

$$= \max\{0, \sum_{i=1}^{n} a_{i,j} x_{\mathrm{OPT},2,i,j} - b\}.$$

**Case 2.** If $\sum_{i=1}^{n} a_{i,j} x_{\mathrm{OPT},2,i,j} - b < 0$, then the smallest feasible value for $y_{\mathrm{OPT},j}$ is

$$y_{\mathrm{OPT},j} = 0$$

$$= \max\{0, \sum_{i=1}^{n} a_{i,j} x_{\mathrm{OPT},2,i,j} - b\}.$$

Therefore, at any global minimum of Definition 4.7, we have

$$y_{\text{OPT},j} = \max\{0, \sum_{i=1}^{n} a_{i,j} x_{\text{OPT},2,i,j} - b\}, \ \forall j \in [m].$$

Substituting $y_{\text{OPT},j}, \forall j \in [m]$ back into the objective of Definition 4.5, we can eliminate the variable $y$ and result in the same problem as Definition 4.5. Thus, we finish the proof. □

## D BASELINE ALGORITHMS FOR THE SOFT AUTHOR NOMINATION LIMIT PROBLEM

In this section, we present two useful baselines for the soft author nomination limit problem described in Section 3.3.

---

**Algorithm 5** Simple Random Algorithm for Desk-Rejection Risk Minimization with Soft Author Nomination Limit Problem in Definition 4.7

---

1: **procedure** RANDASSIGNSOFT($a \in \{0,1\}^{n \times m}, n, m, b \in \mathbb{Z}_+$)
2:     $x \leftarrow \mathbf{0}_{n \times m}$
3:     **for** $i \in [n]$ **do**                   ▷ Iterate all the papers
4:         $S_{\text{all}} \leftarrow \{j \in [m] : a_{i,j} = 1\}$     ▷ A set including all the authors for this paper
5:         $S_{<b} \leftarrow \{j \in S_{\text{all}} : \sum_{i=1}^{n} a_{i,k} x_{i,k} + 1 \le b\}$   ▷ Authors nominated by less than $b$ times
6:         **if** $|S_{<b}| = 0$ **then**              ▷ Increase the penalty term
7:             Randomly choose an element $k$ from $S_{\text{all}}$
8:         **else**                   ▷ No need to increase the penalty term
9:             Randomly choose an element $k$ from $S_{<b}$
10:        **end if**
11:        $x_{i,k} \leftarrow 1$
12:     **end for**
13:     **return** $x$
14: **end procedure**

---

**Algorithm 6** Simple Greedy Algorithm for Desk-Rejection Risk Minimization with Soft Author Nomination Limit (Definition 4.7)

---

1: **procedure** GREEDYASSIGNSOFT($a \in \{0,1\}^{n \times m}, n, m, b \in \mathbb{Z}_+, p \in (0,1)^m$)
2:     $x \leftarrow \mathbf{0}_{n \times m}$
3:     **for** $i \in [n]$ **do**                   ▷ Iterate over all papers
4:         $S_{\text{all}} \leftarrow \{j \in [m] : a_{i,j} = 1\}$     ▷ A set including all the authors for this paper
5:         Define $h(j) = p_j + \lambda \max\{0, \sum_{i=1}^{n} a_{i,j} x_{i,j} + 1 - b\}$   ▷ Cost increase if we assign to author $j$
6:         $S_{\min} \leftarrow \{j \in S_{\text{all}} : h(j) = \min_{k \in S_{\text{all}}} h(k)\}$     ▷ Authors with minimal cost increase
7:         Randomly choose an element $k$ from $S_{\min}$
8:         $x_{i,k} \leftarrow 1$
9:     **end for**
10:     **return** $x$
11: **end procedure**

---

## LLM USAGE DISCLOSURE

LLMs were used only to polish language, such as grammar and wording. These models did not contribute to idea creation or writing, and the authors take full responsibility for this paper's content.

