# OpenReview forum: "Which Coauthor Should I Nominate in My 99 ICLR Submissions? A Mathematical Analysis of the ICLR 2026 Reciprocal Reviewer Nomination Policy"
_ICLR.cc/2026/Conference — ICLR 2026 Conference Withdrawn Submission_

### Official Review · Reviewer_pRJR · 2025-10-15

**Soundness:** 4
**Presentation:** 4
**Contribution:** 1
**Rating:** 2
**Confidence:** 4

**Summary:**

This paper addresses the problem of minimizing desk rejections in AI conferences that have implemented a reviewer nomination policy. Under this policy, each submission must nominate one of its authors as a reviewer, and if that author is deemed irresponsible, the paper is desk-rejected. The authors investigate how to optimally nominate reviewers to minimize the risk of desk rejections, assuming each author has a certain probability of being irresponsible. Then the authors formalize three variants of the desk-rejection risk minimization problem: a basic problem minimizing expected desk rejections, a hard nomination limit variant restricting how many papers can nominate the same author, and a soft nomination limit variant allowing some flexibility in nominations. They provide efficient algorithms for each variant, including a greedy algorithm for the basic problem, a minimum-cost flow approach for the hard limit problem, and a linear programming solution for the soft limit problem.

**Strengths:**

- The paper offers a novel perspective on strategically nominating reviewers under new conference policies to minimize desk rejections, which is an interesting conceptual note.
- The paper is well-written and clearly structured, making it easy to follow the problem formulations and proposed solutions. The authors not only present the problems and solutions, but also provide examples for some of the claims, making the paper accessible to non-technical readers.

**Weaknesses:**

- The modeling of the problem has a weak motivation. While the basic problem (Definition 3.1) is naturally motivated, in practice, there are no hard or soft nomination limits (Definitions 3.4 and 3.9) imposed by conferences. The authors motivate these variants as ways to prevent widespread failures if one author is irresponsible, but this seems contrived. As the expected number of desk rejections is already minimized in the basic problem, it is unclear why one would want to impose additional constraints that could potentially increase the number of desk rejections.
- All three variants of the problem are joint optimization problems over all authors to minimize the total number of (expected) desk rejections. However, in reality, the authors do not have the incentive to cooperate with each other to minimize the total number of desk rejections. It is unclear which party would implement the proposed algorithms.
- Relatedly, the practical impact of this work is limited. The paper does not provide any empirical evaluation or real-world case analysis to demonstrate the effectiveness of the proposed algorithms. Without such evidence, it is difficult to assess the practical utility of the methods.
- Technically, the paper reads like a juxtaposition of three sets of problem formulations and their solutions using very basic techniques. None of the solutions (e.g., greedy algorithm, minimum-cost flow, linear programming) introduce novel technical insights that would be of interest to the broader research community that studies optimization problems.

**Questions:**

Feel free to address any of the weaknesses above.

---

> ### Author Response · Authors · 2025-11-23
>
> We are grateful for your review and the helpful points you raised. Thank you for your support. We will address all of them in the next version.

---

### Official Review · Reviewer_FitC · 2025-10-20

**Soundness:** 2
**Presentation:** 3
**Contribution:** 1
**Rating:** 2
**Confidence:** 5

**Summary:**

This paper studies an optimization problem in peer review called Desk-Rejection Risk Minimization:
In peer review, each paper nominates an author as the reviewer. Each author $i$ has a probability $p_i$ of being irresponsible. The goal is for a central coordinator to assign nominations so that the expected number of desk-rejected papers (due to irresponsible reviewers) is minimized.

The paper shows:

* When each author can be nominated by multiple papers, a simple greedy algorithm is optimal.

* When each author can be nominated by at most k papers, the optimal solution can be obtained by a minimum-cost flow algorithm.

* When the constraint in (2) is soft, a feasible solution can be obtained via LP rounding.

**Strengths:**

The problem is natural and novel.
The paper is mostly well written, and the examples are clear and helpful.

**Weaknesses:**

The theoretical contribution is weak.
(1) The greedy algorithm’s optimality in the unrestricted case is trivial.
(2) The use of (bounded) min-cost flow to solve the problem in Definition 3.4 is natural and already well known.
Moreover, the paper’s claim that the problem in Definition 3.4 is equivalent to min-cost flow is incorrect.
After checking the appendix, I believe the proof only shows that the problem can be reduced to min-cost flow, but not the other direction. In fact, I do not think the reverse statement is true.
(3) The LP rounding method in the soft-constrained case is standard and lacks any approximation bound.

The assumptions and setting are not realistic. The paper minimizes the number of desk-rejected papers from a centralized viewpoint, but maximizing author welfare is not the objective of any real agent, including conference organizers. The assumption that each author’s irresponsibility probability is fixed and known is also unrealistic.

Since peer review is inherently a decentralized system, studying the Price of Anarchy might be more meaningful than focusing on risk minimization.

**Questions:**

The topic of “desk rejection caused by irresponsible reviewers” is interesting and meaningful. The authors could extend their work by exploring broader objectives, considering strategic behaviors, or analyzing the Price of Anarchy in such systems.

---

> ### Author Response · Authors · 2025-11-23
>
> We appreciate your constructive suggestions and careful review. Thank you for helping us improve our manuscript. We will incorporate these points in the next version.

---

### Official Review · Reviewer_pcpH · 2025-11-01

**Soundness:** 1
**Presentation:** 2
**Contribution:** 1
**Rating:** 0
**Confidence:** 5

**Summary:**

This paper studies the recently introduced ICLR 2026 reviewer nomination policy, under which each submission must designate one of its authors as a reviewer, and a paper may be desk-rejected if its nominated reviewer performs irresponsibly. The authors formalize this situation as an optimization problem where each author has a fixed “irresponsibility probability,” and each paper must nominate exactly one of its authors subject to optional limits on how many nominations any author can accept.

Three problem variants are proposed:

Basic expected-risk minimization (Def. 3.1) — solved by a greedy assignment.

Hard nomination limits (Def. 3.4) — formulated as a constrained linear program and claimed equivalent to a minimum-cost flow problem (Theorem 4.3).

Soft nomination limits (Def. 3.9) — formulated as a convex program with linearization and rounding.

The paper claims novelty in providing the first theoretical framework for reciprocal reviewer nomination policies and argues that these formulations could inform authors’ decisions in real submission systems.

**Strengths:**

Clear mathematical exposition and formal structure.

Connection between linear programming and minimum-cost flow is correctly stated (though trivial).

This is a timely reference to an ongoing change in ICLR conference policy, and the field at large.

**Weaknesses:**

1. *Lack of substantive motivation*

The introduction (L35–42) devotes a full page to the “rise of AI" and the role of ML conferences, which is unrelated to the specific optimization problem, and tangential to the problem setting.

The scenario itself is implausible: no author submits hundreds of papers to a conference, and reciprocal review caps are typically not much smaller (if at all) than reviewer submission counts.

The claimed problem of “which coauthor to nominate” is neither pressing nor grounded in evidence that authors systematically mis-nominate reviewers.

2. *Unrealistic and ill-posed modeling assumptions*

Assigning a fixed irresponsibility probability $p_j$ to each author (L67) is sociologically and operationally meaningless; there is no mechanism to estimate these probabilities.

The model ignores the structure of research groups, institutional roles, or the matching mechanisms already used by conference chairs.

The “limit b” on the number of nominations per author (Def. 3.4) is not motivated—its nontrivial unless b < number of submissions, and how many people are submitting enough papers for this to make much difference?


3. *Technical triviality*

The “basic” problem (Def. 3.1 → Prop. 3.3) is separable and solved by a one-line greedy rule.

The “hard limit” version is an immediate instance of a capacitated bipartite matching / transportation problem, well known since the 1950s.

The “soft limit” version is just a convex relaxation with standard linearization; no new method or proof technique appears.

No empirical evaluation, simulation, or complexity analysis beyond textbook bounds.

4. *Misstatement (or extreme misleadingness) of Proposition 4.2*

Prop. 4.2 asserts the existence of fractional optima in the LP relaxation. While true in a narrow sense (fractional can occur in degenerate cases), the text misleadingly suggests that the LP relaxation is generally non-integral.

In fact, the constraint matrix (each paper nominates one author; each author ≤ b papers) is totally unimodular. Hence every vertex of the feasible polytope is integral, and an integral optimum always exists unless costs are exactly tied.

Thus, Theorem 4.3’s appeal to “minimum-cost flow integrality” adds little: integrality was already guaranteed by very basic linear programming theory.

Suggested correction: clarify that fractional optima only arise under tie conditions, and that the LP already guarantees an integral optimal vertex.

5. *Conceptual disconnect from ML venue*

The paper contains no learning, no data, and no connection to representation, optimization, or generalization—its content is pure combinatorial optimization.

The “ICLR policy” context is not sufficient to justify ICLR as a venue; the paper would at best fit a workshop on peer review systems or theoretical computer science.

The tone (“We design efficient, principled nomination strategies…”) exaggerates novelty for a standard LP→flow mapping.

**Questions:**

Clarification on modeling assumptions:

The paper assumes that each author submits multiple papers and can nominate reviewers reciprocally. Can the authors provide empirical evidence that this setting reflects real submission behavior in modern ML conferences?

Scalability and practicality:

How do the authors envision their proposed optimization model being implemented in a real review assignment pipeline? Who would compute or maintain the required parameters (e.g., probabilities, quotas)?

Relation to prior work:

The authors briefly mention existing reviewer assignment algorithms. Could they clarify what is novel about their formulation beyond classical linear assignment models?

Relevance to ICLR:

Given that the paper focuses on a combinatorial optimization formulation unrelated to learning or neural networks, could the authors explain why this is an appropriate venue for the work?

Experimental validation:

The paper contains no experiments or simulations. Have the authors considered evaluating how their approach compares to real-world reviewer assignment data (e.g., OpenReview statistics)?

---

> ### Author Response · Authors · 2025-11-23
>
> We sincerely thank you for your time and effort in reviewing our work. Your insights are valuable to us. We will carefully address them in the next version.

---

### Official Review · Reviewer_81tn · 2025-11-04

**Soundness:** 2
**Presentation:** 3
**Contribution:** 2
**Rating:** 2
**Confidence:** 4

**Summary:**

This paper considers the problem that assuming each author carries a probability of being irresponsible, we ask: how can authors (or automated systems) nominate reviewers to minimize the risk of desk rejections? Formalizing it as a discrete optimization (with soft or hard constraints), the paper introduces an efficient solution based on min-cost flow algorithm.

**Strengths:**

This paper considers an interesting problem in peer review and connects it to the classic discrete optimization. Moreover, its solutions can be adapted to several natural variants of the problem.

**Weaknesses:**

The entire framework hinges on a massive, unaddressed assumption: that authors have access to the irresponsibility probability for all their co-authors. The paper provides no guidance on how to estimate or acquire this data, which is the most critical input to all the proposed algorithms.

Moreover, the motivation of this study is a bit shaky in the sense that we want to penalize more responsible reviewers as they are more likely to be nominated as the reciprocal reviewer for the sake of reducing desk rejection rate.

**Questions:**

How can we resolve the incentive issue here that the reviewers want to pretend to be less responsible so that they are less likely to be nominated?

---

> ### Author Response · Authors · 2025-11-23
>
> Thank you for your thoughtful feedback. Your comments are very helpful and much appreciated. We will address these in the next version.

---

### Note · Authors · 2025-11-23

**Comment:**

We would like to sincerely thank all the reviewers for providing insightful feedback. After careful consideration, we have decided to withdraw this paper.

**Withdrawal Confirmation:**

I have read and agree with the venue's withdrawal policy on behalf of myself and my co-authors.